# Is Sustainable Consumption Translated into Ethical Consumer Behavior?

**Monica-Maria Tomșa, Andreea-Ioana Romonți-Maniu and Mircea-Andrei Scridon \***

Faculty of Economics and Business Administration, Babeș-Bolyai University, 400591 Cluj-Napoca, Romania; monica.tomsa@econ.ubbcluj.ro (M.-M.T.); andreea.maniu@econ.ubbcluj.ro (A.-I.R.-M.)
\* Correspondence: andrei.scridon@econ.ubbcluj.ro

**Abstract:** Nowadays, sustainability is assumed to have high potential for promoting ethical consumer behavior. The aim of this study was to analyze the influence of sustainable behavior on consumer intention to be ethical when it comes to political, social, and environmental dimensions. Therefore, insightful results can be brought forward to explain consumer ethical behavior from a different perspective. Covariance structural equation modelling in AMOS was used for data analysis. Three antecedents, namely environmental, social, and economic dimensions of sustainable consumption, are found to have a significant and positive impact on intention to engage in ethically consumer behavior. In this context, companies seeking to proactively approach eco-friendly consumers will need to understand the complexity of the decision-making process of ethically minded consumers.

**Keywords:** sustainability; ethical consumption; decision-making; ethically minded consumer behavior; eco-friendly consumers





## 1. Introduction

In recent years, sustainable consumption and environmentally ethical behavior have become fashionable or "magic formula" terms used in various contexts by people of different backgrounds. Although popular, they seem to describe the same thing, and little attention is given to the differences between them. Thus, a critical question arises: How is sustainable consumption translated into ethical consumer behavior?

Consumer and government interest in sustainable actions and policies is in an upward trend, e.g., the number of countries covered by e-waste legislation and regulation increased continually from 61 in 2014 to 78 in 2019 [1]. The global e-waste recycling market is predicted to grow from $8.4 billion in 2019 to an estimated value of $20.5 billion by 2027 [1]. Similar tendencies can be identified in other industries, e.g., plastic waste recycling ($34 billion in 2019 and $60 billion estimated for 2027) [2]. At the same time, ethical consumption is increasing considerably [3]. For example, the global market value of ethically labelled food products is set to increase from a value of $793.8 billion in 2015 to an estimated value of $900 billion in 2021 [4]. A similar direction is observed for revenues of Fairtrade international products, which increased from $5.5 billion in 2013 to $9.8 billion in 2018 [4]. Therefore, more and more businesses realize the need to consider ecological and human welfare implications when adopting sustainable principles.

Prior research has examined aspects such as social ethical issues [5], environmentally ethical behavior [6], and ethical policy implications [7]. Furthermore, the traditional components of sustainability are explored in a considerable number of studies, offering, therefore, a thorough understanding of the subject. Nevertheless, the research literature is largely silent regarding the link between sustainable consumption practices and environmentally ethical behavior.

Therefore, the present study explores the following research questions: (1) How does the environment component of sustainable consumption influence the ethical consumer behavior at the political, social, and environmental level? (2) Does the social dimension

of sustainable consumption have an impact on ethical social behavior? (3) Is there a relationship between economic aspects of sustainable consumption and environmentally ethical behavior? Our empirical analysis, based on data collected from 332 individuals engaging both in intended sustainable consumption and ethical behavior (stated), explores the mechanisms through which sustainable consumers' actions affect their ethical practices.

Our study contributes to the sustainable and ethical research literature in several ways. This research is the first to focus on the relationship between sustainable and ethical behavior both measured at the component level. We show how the environmental facet of sustainability drives consumer ethical behavior expressed at political and environmental levels. We demonstrate that if consumers manifest interest in social sustainable consumption, they usually adopt an avoidance behavior towards brands and companies that do not respect their employees' basic human rights. We also shed light on the complex relationship between economic aspects of sustainability and environmentally ethical intentions. Thus, this study offers a comprehensive examination of the link between sustainability and ethical behavior.

The article begins with a theoretical approach relating to sustainable consumption and ethical consumer behavior. Then, a conceptual framework is outlined based on the research questions. This framework was used to guide empirical survey research demonstrating the influence of sustainable practices on ethical actions adopted by consumers. Our study ends with a section dedicated to discussion and conclusions, with limitations and avenues for future research also being explored.

### 1.1. Theoretical Approach

Certainly, there is a need to consider paradigm shift from conventional consumption habit to sustainable consumption behavior. Consumers also need to take a certain level of responsibility to make this environmental movement stronger [8]. However, if they feel they are sacrificing too much monetarily, green concerns are secondary considerations, and are given less importance [9].

Sustainable consumer behavior has a variety of forms, from environmental friendliness [10], the interest in organic and fair-trade labels in purchase decisions [11], to the willingness of consumers to pay for local food [12] or caring for oneself in a responsible way [13].

It has been shown that consumers are becoming increasingly aware of environmental issues as a dimension of sustainable consumption [14]. Hence, motivated by their individual beliefs, attitudes, perceptions, consciousness, and personal moral obligations both towards others and the environment, consumers may buy in an environmentally friendly manner. However, there is no unique definition of environmentally friendly consumption, but there are some aspects that could define the environmental dimension of sustainable consumption (ESC). These factors refer to concepts such as recycling, packaging, resources and energy, production, and climate [13].

A growing body of research has started to explore sustainable consumer behavior from a social perspective. Hence, the literature approaches this topic as a motivation to consume in a socially responsible manner that is based on the consciousness of doing something good for others [13]. Other studies focus on social innovations as alternative practices or new variations of practices that differ substantially from established or mainstream routines. This new perspective explains that innovative practices must be more than just ideas or experiments [15].

Geiger et al. [16] explored what matters in sustainable consumption. In this research context, the authors demonstrated that fair prices, health issues, and fair distribution represent important determinants of consumer decision-making when it comes to the economic dimension of sustainable consumption.

Another study conceptualized the economic dimension as a "three-dimensional second-order construct" with each facet describing specific aspects of economically sustainable consumption. Moreover, the authors also noted that the strength of the relations with

the environmental and social dimensions differ for voluntary simplicity and debt-free and collaborative consumption. Thus, these three distinct but strongly interrelated constructs can be combined into one dimension of economic sustainable behavior (the economic dimension of sustainable consumption—ECOSC). This three-dimensional approach has been proven to suitably reflect the complex reality of the economic world [13].

Recently, companies have been making efforts to integrate ethics into their overall strategies [17]. Empirical evidence shows that more and more consumers are also attracted by green consumption and ethical values [18]. The ethics of the consumer are visible from his or her choices and actions. They are based on a thoughtful, planned, and conscious approach [9]. Ethical consumption can be analyzed from several perspectives, but the most important are political, social, and green or environmental approaches [17].

The political dimension of ethical consumption can be defined as the willingness of people to be active and to change things. Here, the concepts such as justice and equality among all human beings are crucial. In this context, they buy fair-trade products to improve the living conditions of producers or, conversely, they boycott companies that they consider not to be consistent with their ethical values [17].

The social perspective of ethical behavior can take several forms, including solidarity, sharing, caring for others, altruism, helpfulness, compassion, and generosity. This dimension can be translated into the use of shared products, fair-trade products [11], donations to charities [5], or the purchase of regional products and buying from small farmers.

Ethical consumption is also a way of living together. It creates a social link between individuals, whereas today consumers are pushed to become more individualist. The social dimension of ethical consumption puts solidarity at the heart of people relationships [17].

The environmental ethical consumption is also called ecological behavior. It focuses on consumers' concern for the environment and uncertainty about the future of human life on the planet [17]. There is a need for a change in attitude towards sustainability, with the food sector leading the way by fostering local production and educating consumers on organic alternatives that are available for purchase [9]. Therefore, companies and people are reorienting towards the development of organic agriculture, renewable energy, the search for simplicity in daily life, recycling, etc.

*1.2. Conceptual Framework and Hypotheses*

Sustainable consumption is considered an important aspect in the global campaign towards a more equitable pattern of development to reverse the negative impacts of human activities on the planet [19].

Motivations for ethical consumption are multiple. Sometimes, ethical judgment is not sufficient to explain ethical consumer behavior. In some cases, the ethics of consumers can be seen to oscillate depending on context and opportunities that arise [20]. In consequence, consumers exercise their ethical principles because their sustainable actions are activated. Therefore, this study investigated whether sustainable consumption is always translated into ethical behavior in the context of Romanian consumers. A further aim was to measure the influence of certain dimensions of sustainable behavior on the dimensions of ethical consumption.

The most reliable and used measurement scale for ethical consumer behavior was the Consumer Ethics Scale (CES) from Muncy and Vitell [21], a scale that was improved in 2005 [22]. Despite the willingness of the authors to improve the CES, several criticisms have been noted over time. For instance, in terms of its applicability, some items are no longer relevant today ("recording a movie through television"). Our study used a different approach where consumers are active and their behavior voluntary and supported by ethical motivations. Hence, the items for ethical consumer behavior (Table A1) were adapted from an updated scale [17].

Concerning sustainable consumption, our study used a measurement model based on the classical three-dimensional approach of sustainability, which is in line with the normative triple bottom line concept. Here, the economic dimension of sustainable consumption

(ECOSC) has three distinct but correlated subdimensions. This three-dimensional approach has been proven to suitably reflect the complex reality of the economic world [13].

If individuals are preoccupied with utilizing products that are produced in an environmentally friendly manner and that are made from recycled materials, then it is hypothesized that they have a propensity to buy and/or support entities that provide or offer eco-labelled products [7,23].

Several authors [11,24–26] are also interested in the relationship between environmentally friendly production and fair-trade policies [27,28] or solidarity with producers supporting fair trade. This is important because of issues with some agricultural products (for example coffee, cocoa, tea) or raw materials for the clothing industry (cotton), such as producers from developing countries not being fairly compensated for they effort. Therefore, the following hypothesis was formulated:

**Hypothesis 1 (H1).** *The environmental dimension of sustainable consumption (ESC) is positively related to the political dimension of ethical consumption (PEC).*

Environmentally friendly interests manifested by consumers are reflected in their reluctance to buy from or support companies or brands involved with human rights issues and child labor. A number of studies [9,29,30] investigated the impact of (non)-ecological production in ensuring protection of employee rights.

Furthermore, recycling practices (mostly informal and regulated) for e-waste products have come sharply into focus in recent years because of direct and indirect child labor usage in developing economies [31–33]. Based on this, we formulated hypothesis H2:

**Hypothesis 2 (H2).** *The environmental dimension of sustainable consumption (ESC) is positively related to the social dimension of ethical consumption (SEC).*

The environment component of sustainability could be considered as the root cause for adopting a voluntary simplicity lifestyle [34]. It could be argued that having interests in protecting the environment reflects in behavior oriented towards altering individual actions such as reducing one's consumption of goods and services to only what is really needed [35–37].

In addition, those types of consumers are involved in daily activities geared towards environment conservation and reducing their carbon footprint to contribute less to global warming [6,38,39]. For example, buying "green" or "eco-labelled" products is seen by consumers in various countries (UK, USA, France) as a way to reduce the impact of global warming [39]. Thus, we hypothesized that:

**Hypothesis 3 (H3).** *The environmental dimension of sustainable consumption (ESC) is positively related to the environmental dimension of ethical consumption (EEC).*

According to Balderjahn et al. [13], the social component of sustainable consumption deals mainly with beliefs held by consumers regarding human rights protection, not utilizing illegal child labor, and fair treatment or compensation of employees. Therefore, a connection develops between socially sustainable consumption and social ethical behavior (boycott) adopted by consumers [35,40,41]. It is further amplified when illegal child labor is perceived as being used by companies (for example sweatshops connected to fast fashion or apparel industries) [42–44].

Furthermore, in some cases consumers engage in anti-consumption practices towards multinational companies when they perceive that workers, usually situated in third-world countries, are unfairly paid for their effort, or are treated poorly by their employers [43,44]. Based on these arguments, we expected that:

**Hypothesis 4 (H4).** *The social dimension of sustainable consumption (SSC) is positively related to the social dimension of ethical consumption (SEC).*

Completing discussions focused on the traditional components of sustainability, the economic aspects are viewed, according to Balderjahn et al. [13], as multifaceted, namely voluntary simplicity, debt-free consumption, and collaborative consumption.

Voluntary simplicity [34] is shown, in different studies [35,45,46], to have an influence on individual behavior such as restricting consumption to only what is really needed and buying energy-efficient products.

Collaborative consumption [47] is linked with actions (car-pooling, tool sharing, household item borrowing) that can be viewed as small, sometimes daily, personal contributions with a not so insignificant impact on environment protection and conservation [48–50]. Therefore, a relationship between collaborative consumption and environmentally friendly behavior adopted by consumers is assumed to exist.

Debt-free consumption is seen as an antithesis to overconsumption [46,51] and is also shown to alter consumer behavior towards environment protection [35]. Following the previous discussions, we hypothesized the following:

**Hypothesis 5 (H5).** *The economic dimension of sustainable consumption (ECOSC) is positively related to the environmental dimension of ethical consumption (EEC).*

Thus, bringing together all our previously mentioned research hypotheses, we propose the conceptual framework presented graphically in Figure 1.

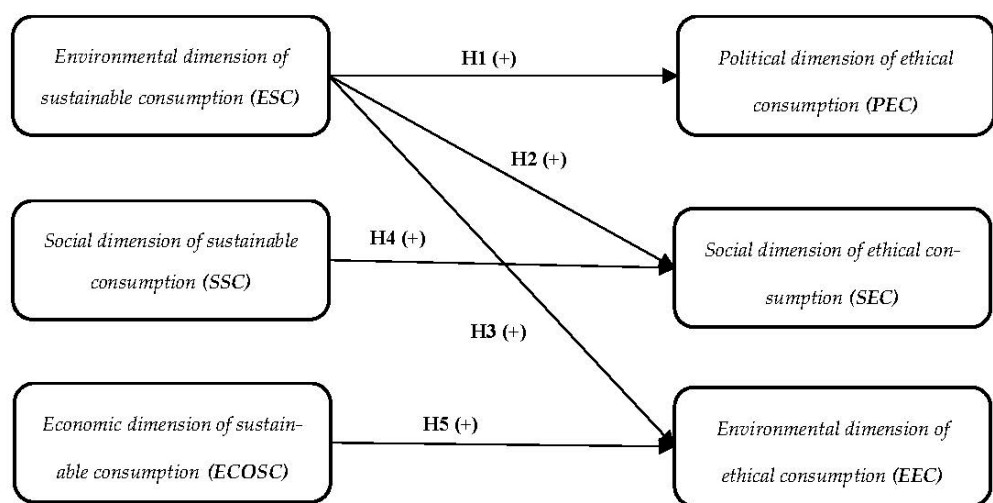

**Figure 1.** Conceptual framework**.**

## 2. Materials and Methods

### 2.1. Measures

All scales used in this study were of a seven-point Likert style. Anchors were either "totally agree" and "totally disagree" or "extremely important" and "extremely unimportant". Both ethical and sustainable behaviors are, in this study, multi-dimensional factors.

The dimensions of ethical behavior are political (PEC), social (SEC), and environmental (EEC). The political and social dimensions each have four items, whereas the environmental dimension is measured using three items. These items were adapted from scales proposed by [17].

Sustainable behavior is also measured at the component level. Scales were adapted from the ones developed by [13]. The environmental component (ESC) was measured using eight items, four for beliefs held and four for the importance of those beliefs for the respondents. The social dimension (SSC) was measured in a similar fashion to the previous one, but these time five items were used both for beliefs held and the importance of those beliefs.

Regarding the third component of sustainable behavior, the economic dimension (ECOSC), due to its complexity, is considered as a second-order latent reflective factor.

The latent factors used in this case were voluntary simplicity, debt-free consumption, and collaborative consumption. Each first-order economic factor was composed of either three (voluntary simplicity and collaborative consumption) or four items (debt-free consumption).

The detailed items for each previously mentioned factor are presented in Tables A1 and A2.

### 2.2. Target Population, Sampling, and Data Collection

The target population was represented by persons from the 18–26 age group, from Romania. This age group was chosen because there is a clear divide between it and older individuals from Romania regarding attitudes towards sustainability and sustainable behavior. The divide can be attributed to enhanced efforts made by Romanian society in the last 25 to 30 years in the direction of encouraging sustainable behavior, after the fall of communism.

Respondents were chosen using judgmental sampling. An online survey was conducted, for two weeks, at the end of February and beginning of March 2020. E-mail lists were compiled with the help of four leading Romanian universities, and at high-school level by engaging school representatives. Thus, we managed to obtain a database with approximately 13,000 entries (e-mails). The questionnaire was prepared in Romanian in order to ensure intelligibility and was distributed as a Google Forms link to selected individuals through e-mail. After eliminating incomplete answers, the response rate was typical for online administered questionnaires, namely 2.7%; thus, we retained responses from 332 individuals.

In the final sample, 84.3% of respondents had at minimum a high-school diploma (12.8% a bachelor's degree, 0.6% a master's degree, and 2.7% were still in the 12th grade), 55.1% of them lived in an urban area, and 48.9% were female, with the average age being 22.6 years (with a sample standard deviation of 2.54 years). Although a non-probabilistic sampling procedure was used, the percentages are approximately equal to those in the target population mentioned earlier (for example, in the 18–26 age group 49.4% of the Romanian population is female, and 52.3% live in an urban area).

### 3. Results

Considering the nature of the nomological model proposed in this paper, covariance structural equation modelling in AMOS was used for data analysis. In the following sections, the analysis and results are presented, starting with exploratory factor analysis, and ending with a summary of the main findings.

### 3.1. Measurement Model

Firstly, in order to evaluate the factors represented in the nomological model, an exploratory factor analysis (EFA) was performed using IBM SPSS. Principal axis factoring, with a Promax rotation, was used for the EFA. Due to poor factor loading, two items from the political dimension of ethical behavior and one from the environmental dimension were dropped, and the EFA was run again. EFA adequacy was evaluated using the KMO statistic (0.909) and Bartlett's test of sphericity ($p < 0.01$).

The number of factors was eight; thus, all first-order factors in the model were retained. The pattern matrix is presented in Appendix B and Cronbach's alpha for each factor in Appendix C.

Secondly, measurement model fit was assessed via confirmatory factor analysis (CFA) in AMOS. No additional items were dropped, resulting in the following fit indices: Cmin/df = 1.945 (df = 306), Comparative fit index (CFI) = 0.964, Goodness-of-fit index (GFI) = 0.884, Adjusted goodness-of-fit index (AGFI) = 0.856, Standardized root mean residual (SRMR) = 0.0568, RMSEA = 0.053, and PClose = 0.186. Based on these indicators, the measurement model fit was deemed adequate, and no further actions were taken for the CFA.

Finally, scale reliability and validity (convergent and discriminant) were investigated. We used the construct reliability index (CR) with values above 0.7 as a threshold for good scale reliability. Convergent validity was assessed using average variance extracted (AVE) for each latent factor, values above 0.5 indicating adequate validity. The results for each component of the model are shown in Table 1.

**Table 1.** Reliability and convergent validity.

|  | **CR** | **AVE** |
|---|---|---|
| EEC | 0.696 | 0.540 |
| PEC | 0.798 | 0.582 |
| SEC | 0.820 | 0.603 |
| ECOSC | 0.721 | 0.573 |
| SSC | 0.970 | 0.865 |
| ESC | 0.958 | 0.849 |

Discriminant validity was investigated using the Fornell and Larker [52] criterion, and no concerns in this regard were found. Complete results are shown in Table 2.

**Table 2.** Discriminant validity.

|  | **EEC** | **PEC** | **SEC** | **ECOSC** | **SSC** | **ESC** |
|---|---|---|---|---|---|---|
| **EEC** | **0.735** | | | | | |
| **PEC** | 0.525 | **0.763** | | | | |
| **SEC** | 0.366 | 0.328 | **0.776** | | | |
| **ECOSC** | 0.438 | 0.183 | 0.360 | **0.757** | | |
| **SSC** | 0.381 | 0.234 | 0.647 | 0.604 | **0.930** | |
| **ESC** | 0.555 | 0.507 | 0.420 | 0.515 | 0.598 | **0.922** |

Considering presented results from the EFA, CFA, and scale reliability and validity, overall good measurement properties are associated with the proposed model. The next section deals with the structural model and hypothesis testing.

### 3.2. Structural Model

Hypotheses were tested using path analysis in the structural model. Table 3 shows whether each proposed relationship between factors is supported or not. It can also be observed that each significant standardized estimate has a positive value. Therefore, the direction—positive influence—is confirmed for H1, H3–H5. The only nonsignificant standardized estimate is associated with H2. Therefore, our data do not support the proposed relationship between environmentally sustainable consumption and socially ethical consumption. The structural model fit statistics are as follows: Cmin/df = 2.035 (df = 313), CFI = 0.960, GFI = 0.877, AGFI = 0.852, SRMR = 0.062, RMSEA = 0.056, and PClose = 0.059. Therefore, no significant differences between the measurement and structural model were observed.

**Table 3.** Results for the structural model.

| Hypothesis | Proposed Relationship | Standardized Estimates | Hypothesis Status |
|---|---|---|---|
| H1 | ESC PEC | 0.507 *** | Confirmed |
| H2 | ESC SEC | 0.007 | Not confirmed |
| H3 | ESC EEC | 0.460 *** | Confirmed |
| H4 | SSC SEC | 0.615 *** | Confirmed |
| H5 | ECOSC EEC | 0.201 ** | Confirmed |

*** $p < 0.01$; ** $p < 0.05$.

The environmental dimension of sustainable consumption positively influences both the political and environmental dimensions of ethical consumption, with a larger impact on the political dimension. On the other hand, the proposed relationship between environmentally sustainable consumption and the social component of ethical consumption was not supported by the available data.

The last two relationships proposed in our model were confirmed by hypothesis testing; therefore, a positive relationship exists between the social component of sustainable consumption and the social dimension of ethical consumption. A positive relationship is also present among the economic dimension of sustainable consumption and the environmental dimension of ethical consumption.

## 4. Discussion and Conclusions

The aim of this study was to analyze the influence of sustainable behavior on consumers intention to be ethical when it comes to political, social, and environmental dimensions. Therefore, insightful results can be brought forward to explain consumer ethical behavior from a different perspective.

First, in the proposed nomological model, a set of five hypotheses was formulated, and results show support for four of them. Therefore, the paths between the dimension of environmentally sustainable consumption and the dimensions of political and environmental ethical behaviors are statistically significant. The relationship between social sustainable consumption and social ethical behavior is supported by the results of this study. Moreover, Hypothesis 5 was confirmed; thus, economically sustainable consumption has a positive impact on environmental ethical consumption.

Second, it is important to note that, while the study did not show environmentally sustainable consumption to have any significant impact on socially ethical consumption, the findings should not be seen to ignore the importance of the relationship supposed to exist between environmentally sustainable consumption and social ethical behavior. On the contrary, this relationship should theoretically exist based on the same arguments as the relationships between environmentally sustainable consumption and political ethical behavior or environmental ethical behavior.

Overall, key implications for both researchers and practitioners include different intentions to engage in ethical behavior, based on the dimensions of sustainable consumption from the proposed model. One significant suggestion is that the decision-making process of environmentally minded consumers is strongly correlated with the intention to adopt a politically ethical behavior. Thus, if individuals attribute a high degree of importance to buying products made from recycled materials or packaging that are produced in an environmentally friendly manner are more inclined to buy products with an eco-label or to shop in stores that promote fair trade [11]. Our findings are similar to results from Zerbini et al. [26], which support fair-trade arrangements and premium payment for supporting producers and workers.

Therefore, such consumers are potentially very attractive to companies because they are willing to engage in positive word-of-mouth for preferred brands and also command higher profits because they usually are able to pay more for eco-friendly products. These results are in line with the study of Maaya et al. [11], which empirically showed that individuals are willing to pay a premium for eco-labelled products. Moreover, these products are indistinguishable from fair-trade goods, in the mind of our respondents, conflicting with van Herpen et al. [27] who argue that premium prices are detrimental to the sales of fair-trade products.

In a similar vein, environmentally minded consumers also engage in actions pertaining to environmental ethical behavior. For example, in a similar fashion to Whitmarsh et al. [37], if individuals are sensitive to issues related to recycling, eco-packaging, and eco-friendly disposal of products, they are more likely to restrict their consumption (food, energy, clothing) to only what it is needed. Thus, comparing results with Roser-Renouf et al. [39] we both show that, in doing so, consumers contribute to the reduction of global warming and

in the medium to long term to the preservation of the environment for future generations. On the other hand, practitioners should not overlook the fact that a more restrictive consumption implies willingness to pay a premium for purchased goods, thus generating higher profit margins.

Aligned with results from Maxwell-Smith et al. [42] and Yoon et al. [44], our findings also highlight that if consumers are interested in human rights protection efforts such as fighting discrimination, illegal child labor, workplace abuse, and unfair treatment of workers, they are more prone to boycotting brands that do not consider protecting basic human rights. Although the negative perspective is dominant, a positive side might also be present in situations involving companies with a proven track record for respecting their workers' rights. In these cases, consumers could become promoters for those companies by engaging in favorable word-of-mouth.

Another key implication emphasizes the complexity of the economic dimension of sustainability (which in this study is treated as a second-order reflective factor) and its various interactions with environmental ethical behavior. Huttel et al. [46] point out why economically sustainable consumption patterns are usually related to making sacrifices. Similarly, we use debt-free and collaborative consumption as examples of economically sustainable consumption and show their significant impact on persons adopting a voluntary restriction behavior when purchasing goods and services. These conflicts with results of Iwata [45], who argues the existence of a low correlation between voluntary simplicity and environmentally responsible consumerism.

According to Philip et al. [48], even though individuals laud protecting the environment and reducing waste by participating in P2P renting, economic utility is still a primary consideration for this behavior, thus also conflicting with our findings. Therefore, some market segments are more challenging to approach using traditional promotional tools. In other words, marketers should consider alternative communication efforts focusing on events, membership, and one-to-one marketing.

*Limitations and Future Research*

In the previous paragraphs, several relevant implications for researchers are presented, but avenues for improving this study should be explored based on some limitations. First, our research should also be conducted in other countries with different cultures, because it did not address the Romanian national culture. According to Hofstede [53], Romania has a different cultural background compared to its neighbors. For example, Romania scores higher on the collectivism and femininity dimensions than surrounding countries.

Second, similar studies should include actual behavior for both sustainable and ethical dimensions. Such a recommendation is based on the fact that only the intention to engage in these behaviors is actually measured in our study, and it becomes problematic when unethical actions are changed by situational factors.

Third, future studies might include other factors that could play a moderating effect on the relationships presented in the model. Sociodemographic variables like gender, age, income level, and level of education could be examined in this regard. Attitudes towards sustainability might be explored as well as their role as antecedents for both sustainable and ethical behaviors.

Fourth, data were collected using a judgmental sample procedure; therefore, great care must be taken when trying to generalize the results from this study to the entire target population. Therefore, a probabilistic sample procedure, for example cluster sampling, is considered more appropriate to use in this type of study, but it might involve more financial resources.

Fifth, we acknowledge that the triple bottom line approach to sustainability can be improved, along with our proposed model, by adding, for example, governance as a component [54] to better align the goals of key decision-makers at the governmental level with those of citizens. Thus, a more practical set of recommendations can be brought forward in order to assess government policies towards sustainability, e.g., by using scorecards [54].

**Author Contributions:** Conceptualization, M.-M.T. and M.-A.S.; methodology, M.-A.S.; software, M.-A.S.; validation, M.-M.T., A.-I.R.-M. and M.-A.S.; formal analysis, M.-A.S.; investigation, M.-M.T., A.-I.R.-M.; resources, A.-I.R.-M.; data curation, A.-I.R.-M.; writing—original draft preparation, M.-M.T. and M.-A.S.; writing—review and editing, M.-M.T., A.-I.R.-M. and M.-A.S.; funding acquisition, M.-M.T. All authors have read and agreed to the published version of the manuscript.

**Funding:** This research was funded by Babeş-Bolyai University, grant number GS-UBB-FSEGA-tomsamonicamaria.

**Institutional Review Board Statement:** Ethical review and approval were waived for this study, due to the fact that it was not a medical one, subjects were not exposed to any harm, their rights were protected, and every precaution was taken in order to protect the privacy of research subjects and the confidentiality of their personal information.

**Informed Consent Statement:** Informed consent was obtained from all subjects involved in the study.

**Data Availability Statement:** Data available upon request from the correspondence author.

**Conflicts of Interest:** The authors declare no conflict of interest.

## Appendix A

**Table A1.** Items of the three-dimensional scale of ethical consumption behavior (Likert type scale: 1—totally disagree, 7—totally agree).

| Dimensions | Items | Mean | Std. dev. |
|---|---|---|---|
| *Political Dimension (PEC)* | I prefer buying products with an eco-label (PEC1). | 4.59 | 1.51 |
| | I prefer to buy in shops that highlight the ecological or organic products (PEC2). | 4.46 | 1.58 |
| | I prefer to do my shopping in stores that promote fair trade (PEC3). | 5.27 | 1.40 |
| | I buy fair-trade products in solidarity with producers (PEC4). | 4.57 | 1.51 |
| | I buy products sold through social actions (PEC5). | 4.46 | 1.64 |
| *Social Dimension (SEC)* | I avoid brands/products that profit from the misery of their employees (SEC1). | 4.89 | 1.84 |
| | I avoid products or brands that make children work even indirectly (SEC2). | 4.92 | 1.84 |
| | I avoid products from companies that do not respect the rights of their employees (SEC3). | 5.00 | 1.66 |
| *Environmental Dimension (EEC)* | I restrict my consumption (food, energy, clothing, etc.) to what I really need (EEC1). | 4.56 | 1.73 |
| | I contribute to the preservation of the environment through everyday actions (EEC2). | 5.62 | 1.35 |
| | To reduce my contribution to global warming, I consume differently (EEC3). | 4.63 | 1.45 |

Adapted from [17].

**Table A2.** Consciousness for sustainable consumption scale (Likert type scale: 1—totally disagree, 7—totally agree or 1—extremely unimportant, 7—extremely important).

| Dimensions | Items |
|---|---|
| | *Belief: I buy a product only if I believe that (during the manufacturing) . . .* |
| | × |
| | *Importance: How important is it for you personally that (during the manufacturing of a product) . . .* |
| *Environmental Dimension (ESC)* | It is made from recycled materials? (ESC1) |
| | It can be disposed of in an environmentally friendly manner? (ESC2) |
| | It is packaged in an environmentally friendly manner? (ESC3) |
| | It is produced in an environmentally manner? (ESC4) |
| | Workers' human rights are adhered to? (SSC1) |
| | No illegal child labor is involved? (SSC2) |
| *Social Dimension (SSC)* | Workers are not discriminated against? (SSC3) |
| | Workers are not abused? (SSC4) |
| | Workers are treated fairly or are fairly compensated? (SSC5) |
| | *Belief: Even if I can financially afford a product I buy a product only if I believe that . . . [a]* |
| | × |
| | *Importance: Even if you can financially afford a product, how important is it for you personally that . . . [a]* |
| *Economic Dimension—Voluntary simplicity (ECOSC1)* | I/you really need this product? (ECOSC11) |
| | It is a useful product? (ECOSC12) |
| | I/you absolutely require this product? (ECOSC13) |
| | I/you don't become overindebted in the long term? (ECOSC21) |
| *Economic Dimension—Debt-free consumption (ECOSC2)* | The expenses don't unduly burden my/your financial situation? (ECOSC22) |
| | I/you don't have to forego future purchases? (ECOSC23) |
| | I/you don't have to take money from my/your financial reserve for emergency cases for it? (ECOSC24) |
| | I/you don't want to borrow it from friends? (ECOSC31) |
| *Economic Dimension—Collaborative consumption (ECOSC3)* | I/you really need to own it and don't want to share with others? (ECOSC32) |
| | I/you don't want to rent or lease it? (ECOSC33) |

[a] Divergent wording for items of the economic dimension. Adapted from [13].

**Appendix B**

**Table A3.** Factor pattern matrix.

| | | | | | | | | |
|---|---|---|---|---|---|---|---|---|
| **Factor Pattern Matrix** | | | | | | | | |
| | **Factor** | | | | | | | |
| | **1** | **2** | **3** | **4** | **5** | **6** | **7** | **8** |
| PEC1 | | | | 0.727 | | | | |
| PEC2 | | | | 0.786 | | | | |
| PEC3 | | | | 0.680 | | | | |
| SEC1 | | | | | | | 0.698 | |
| SEC2 | | | | | | | 0.901 | |
| SEC3 | | | | | | | 0.600 | |
| EEC2 | | | | | | | | 0.570 |
| EEC3 | | | | | | | | 0.811 |
| ESC1 | | 0.837 | | | | | | |

**Table A3.** *Cont.*

| | **Factor Pattern Matrix** | | | | | | | |
| | **Factor** | | | | | | | |
| | **1** | **2** | **3** | **4** | **5** | **6** | **7** | **8** |
| ESC2 | | 0.906 | | | | | | |
| ESC3 | | 0.922 | | | | | | |
| ESC4 | | 0.890 | | | | | | |
| SSC1 | 0.823 | | | | | | | |
| SSC2 | 0.768 | | | | | | | |
| SSC3 | 0.973 | | | | | | | |
| SSC4 | 0.952 | | | | | | | |
| SSC5 | 0.958 | | | | | | | |
| ECOSC11 | | | | | 0.958 | | | |
| ECOSC12 | | | | | 0.821 | | | |
| ECOSC13 | | | | | 0.888 | | | |
| ECOSC21 | | | 0.818 | | | | | |
| ECOSC22 | | | 0.975 | | | | | |
| ECOSC23 | | | 0.857 | | | | | |
| ECOSC24 | | | 0.865 | | | | | |
| ECOSC31 | | | | | | 0.825 | | |
| ECOSC32 | | | | | | 0.890 | | |
| ECOSC33 | | | | | | 0.831 | | |

Extraction method: principal axis factoring. Rotation method: Promax with Kaiser normalization.

**Appendix C**

**Table A4.** Scale reliability.

| Dimension | Number of Items | Cronbach's Alpha |
|---|---|---|
| PEC | 3 | 0.774 |
| SEC | 3 | 0.821 |
| EEC | 2 | 0.629 |
| ESC | 4 | 0.957 |
| SSC | 5 | 0.968 |
| ECOSC1 | 3 | 0.932 |
| ECOSC2 | 4 | 0.935 |
| ECOSC3 | 3 | 0.884 |

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
