# Peer review of "Is Sustainable Consumption Translated into Ethical Consumer Behavior?"

_sustainability, doi:10.3390/su13063466_

Round 1

Reviewer 1 Report

The authors try to use surveys to answer the question of how different sustainability dimensions will affect consumers' consumption behaviors. I have the following concerns.

  1. The survey questions are misleading and biased (appendix A). The participants are more likely to answer "yes" to most of your questions.
  2. It is unclear how your different dimensions of sustainability are quantified and measured.
  3. Why do you choose the current model? Why is it useful and valid?
  4. Please justify your hypotheses.
  5. Please put your model description under the "materials and methods: section and describe your results separately.
  6. Please re-structure your introduction section. It is uncommon to have lots of subtitles under the introduction section. Please fully motivate your research topic and summarizes your research innovations, procedures, and findings in the introduction section.
  7. The English writing requires extensive edits and significant improvements.

Reviewer 2 Report

Dear Authors,

the article is at a fairly early stage of development. This might be an interesting paper which has potential to be published in “Sustainability” journal but it needs some important improvements. I provide some suggestions:

1) Title: In my opinion, the use of the word "always" is unfortunate in the title because it is hard to prove using research methods. However, I understand that the authors may have used this word perversely to emphasize that, unfortunately, ethical principles are often not a priority in practice.

2) Abstract: The aim of the paper should be more precisely defined (should be copied from line 257-260).

3) Introduction: Introduction is too long. Too many aspects have been covered in this section and the structure of the article is complicated. The line 23-75 are clear but 2-3 sentences of summary would be advisable which would emphasize the need for research in this area.

4) Without a doubt, the analysis of the literature in this article is poor and should be well developed.

5) Materials and methods: There is no precise information about the research sample. The description of the research methods is insufficient.

6) Results: Rather properly developed. I have no comments.

7) Discussion and conclusions: The biggest drawback of the paper is the lack of a real discussion and comparison of the results of the authors' research with the research results of other scientists. However, the conclusions are very well developed.

Reviewer 3 Report

The paper has potential and it can be a good contribution to the literature. The authors presents the results of an empirical research which argues that some dimensions of sustainable consumption behaviour could explain ethically practices, in terms of justice and equality among all human beings (political dimension), solidarity at the heart of people relationships (social dimension) and consumers’ concern for the environment and uncertainty about the future of human life on the planet (ecological or environmental dimension). Nevertheless, some aspects need to be changed or better explained before being considered for publication, for example:

  • The abstract could be improved, for example including more achievements;
  • The introduction can be improved, for example emphasizing the diverse aspects that are related to sustainability (see Cruz and Marques, 2014);
  • Highlight the novelty of this study in the introduction;
  • Moreover, I would recommend to move the text between line 34 and 160 to a new chapter that could be named as “Theoretical approach”
  • A paragraph presenting the organization of the paper should be included in the end of the introduction;
  • Literature review can be improved;
  • All the abbreviations must be presented in the text;
  • Regarding the methodology, the model must be better justified, including the limitations;
  • Explain better the Figure 1;
  • The authors could improve the discussion providing more insights about the future with this approach;
  • Try to explain better the significance of the statistical results in table 3 and the meaning of the sign of the variables;
  • The paper would benefit with a final table summarizing the achievements per hypothesis analyzed;
  • More recommendations for the decision makers were expected in the conclusions;
  • The references must be homogenized and in line with the author guidelines (for example, some issues are missing).

References:

CRUZ, N.; MARQUES, R. (2014). Scorecards for sustainable local governments. Cities. Elsevier. ISSN: 0264-2751. Vol. 39, pp. 165–170.

Round 2

Reviewer 1 Report

Thanks for the authors' efforts in addressing all my concerns.

Reviewer 2 Report

Dear authors,
thank you for the reconstruction of the article and the thorough verification and supplementation of the paper content according to the indicated comments. The quality of the article is much better. For this reason, I recommend the manuscript to be published in the journal.

Reviewer 3 Report

The paper is much better now and it can be accepted.